# Embedding the Patient-Citizen Perspective into an Operational Framework for the Development and the Introduction of New Technologies in Rehabilitation Care: The Smart&Touch-ID Model

**DOI:** 10.3390/healthcare11111604

**Published:** 2023-05-30

**Authors:** Olivia Realdon, Roberta Adorni, Davide Ginelli, Daniela Micucci, Valeria Blasi, Daniele Bellavia, Fabrizio Schettini, Roberto Carradore, Pietro Polsinelli, Marco D’Addario, Marco Gui, Vincenzina Messina, Emanuela Foglia, Patrizia Steca, Fabrizia Mantovani, Francesca Baglio

**Affiliations:** 1Department of Human Sciences for Education, University of Milano-Bicocca, 20126 Milan, Italy; 2Department of Psychology, University of Milano-Bicocca, 20126 Milan, Italy; 3Department of Informatics, Systems, and Communication, University of Milano-Bicocca, 20126 Milan, Italy; 4IRCCS Fondazione Don Carlo Gnocchi ONLUS, 20148 Milan, Italy; 5Centre for Health Economics, Social and Health Care Management, LIUC-Università Carlo Cattaneo, 20153 Castellanza (VA), Italy; 6Department of Sociology and Social Research, University of Milano-Bicocca, 20126 Milan, Italy; 7Open Lab srl, 50121 Florence, Italy

**Keywords:** telerehabilitation, noncommunicable diseases, needs assessment, technology assessment, psychological well-being, community-based participatory research, co-design, multidimensional approach

## Abstract

To date, at least 2.41 billion people with Non-Communicable Diseases (NCDs) are in need of rehabilitation. Rehabilitation care through innovative technologies is the ideal candidate to reach all people with NCDs in need. To obtain these innovative solutions available in the public health system calls for a rigorous multidimensional evaluation that, with an articulated approach, is carried out through the Health Technology Assessment (HTA) methodology. In this context, the aim of the present paper is to illustrate how the Smart&TouchID (STID) model addresses the need to incorporate patients’ evaluations into a multidimensional technology assessment framework by presenting a feasibility study of model application with regard to the rehabilitation experiences of people living with NCDs. After sketching out the STID model’s vision and operational process, preliminary evidence on the experiences and attitudes of patients and citizens on rehabilitation care will be described and discussed, showing how they operate, enabling the co-design of technological solutions with a multi-stakeholder approach. Implications for public health are discussed including the view on the STID model as a tool to be integrated into public health governance strategies aimed at tuning the agenda-setting of innovation in rehabilitation care through a participatory methodology.

## 1. Introduction

To date, at least 2.41 billion people with Non-Communicable Diseases (NCDs) are in need of rehabilitation. Rehabilitation treatments through innovative technologies that can make this service widespread, up to the patient’s home, are showing growing substantial evidence of their effectiveness (see, for instance, [1]).

To obtain these solutions available in the public health system calls for a rigorous multidimensional evaluation that, as a complete and valid approach, is developed through the implementation of the Health Technology Assessment (HTA) methodology. However, the route to integrate patients’ perspectives of evaluation, considered essential from this standpoint, is still much debated (see, for instance, [2]). Moreover, the multidimensional evaluation framework can be applied at different stages of the development of a technology, ranging from the very first conceptualizations (with the early assessment approach) to existing prototypes virtually ready for introduction and testing in the market and in the healthcare system.

The aim of the present paper is, therefore, to illustrate how the Smart&TouchID (STID) model addresses the need to incorporate patients’ evaluations into a multidimensional technology assessment framework by presenting a feasibility study of model application with regard to rehabilitation experiences of people living with NCDs.

Although the STID model operates to optimize the co-design and development of innovative solutions for rehabilitation care for NCDs, in the present paper the focus is not on specific technologies as outputs of the STID multidimensional evaluation flow. Rather, the attention is drawn to the processes that the STID model enables in order to integrate the standpoints (action and attitudes) of the various stakeholders to obtain technology optimization. This way, the paper targets the feasibility testing of the STID model as a tool in national/regional governance strategies aimed at tuning the agenda-setting of innovation in rehabilitation care through a participatory methodology.

After detailing how technological innovation can address the growing need for rehabilitation of people with NCDs, an overview of the STID model will be illustrated, showing how it models the development of health technologies for people with NCDs into a flow that embeds the evaluation of digital solutions from the very first stages of their development and from the perspectives of different stakeholders including patients. Then, as a feasibility study in running the STID model with regard to the embedding of patients’ and citizens’ evaluations in technological development, preliminary evidence from the collection of patients’ and citizens’ experiences on rehabilitation will be described and discussed.

### 1.1. Technological Innovation to Meet the Need for Rehabilitation for People with Non-Communicable Diseases

In the last thirty years, the need for rehabilitation for people with Non-Communicable Diseases (NCDs) has increased to involve up to 2.41 billion people. Innovative technologies for remote rehabilitation care can help reach all those in need.

According to the estimates of the Global Burden of Diseases, Injuries, and Risk Factors Study [3], NCDs have, since 1990, become responsible for a notable proportion of the burden due to Years of Life lived with Disability (YLDs). In 2019, together with those connected with injury, YLDs due to NCDs accounted for over 50% of all disease burdens in 11 countries. Drawing on this evidence, the landmark study by Cieza and colleagues [4] estimated that one-third of the world’s population lives with a health condition that would benefit from rehabilitation. This need has increased by 63% in the last thirty years, going from 1.48 billion to 2.41 billion people. The impact on public health expenditure is considerable: just in Europe, for instance, healthcare costs for NCDs are estimated at around 700 billion euros per year [5]. However, so far, rehabilitation is still construed as a very specialized and expensive service, mostly directed at severe disabilities. This way, it cannot guarantee timed and intensive access to people in the early to medium stages of the disease, when, as documented in the literature, early intervention can load on the cognitive reserve and residual skills, with documented slow-down clinical outcomes [6,7,8].

Digital healthcare technologies, conceived as new means for addressing the big healthcare challenges of the 21st century [5] can provide rehabilitation interventions that through telecommunication and information technologies, guarantee the continuity-of-care for NCDs outside the hospital settings [9,10]. These remote rehabilitation interventions, defined as telerehabilitation (TR), require a “double-loop” communication between the clinic and the patient’s home to be in line with face-to-face treatment and to warrant the fundamental clinical actions: assessment, monitoring, and feedback to the patient [11]. While in TR synchronous models, usually delivered through video conferencing devices, the 1:1 setting is the same as in face-to-face interventions, in asynchronous models the clinician’s actions and the patient’s actions are temporally decoupled. The asynchronous modeling option, therefore, moves beyond the conventional 1:1 setting, enabling a one-to-many simultaneous delivery of rehabilitation treatments. Evidence on the efficiency and efficacy of TR is accumulating, both as directed to people with NCDs [11,12,13] and documenting its non-inferiority, with respect to conventional face-to-face treatments [1,11]. Along this line, tech-enabled rehabilitation care can be conceived as the ideal candidate to scale up rehabilitation to reach all people with NCDs in need, providing accessibility to continuity of care outside the hospital through its integration into the health system. The target is to strengthen rehabilitation services at the primary care level, as advocated by Cieza and colleagues [4].

To reach the aim of making tech-enabled rehabilitation a widespread service, ensuring accessibility and quality home-based management of chronic conditions requires rigorous and evidence-based evaluation regarding its safety, efficacy, and sustainability balance. The Health Technology Assessment (HTA) methodology provides a more complete and valid approach for evaluating if and how in a health system, new technologies can promote equitable and quality care with available healthcare resources. However, in the last ten years, HTA agencies have strongly highlighted that the assessment of new technologies should not only include the technical and financial dimensions of new treatments, but also patients’ perspectives on usability, acceptability, and their full impact on everyday care routines. That is, an expansion in breadth and depth of the HTA approach is called for to include in the evaluation process the evidence on the impact of new technologies both on patients’ health and on their well-being and quality of life [2]. Several mechanisms have been identified for this aim (e.g., the meaningful patient involvement approach, [14]), which encompasses the alignment of commitment of stakeholders (including patients besides clinicians, industries and experts) to shared goals, together with in-depth accounts of patients’ lived experiences to add up to organizational evaluations [15]. Notwithstanding the great variability and an absence of comprehensive, robust practices for patient engagement, the mechanisms identified in the literature essentially call for a change in the way patient evaluations can be embedded, and transform the organizational setup (when, what, and how) of evaluations of all relevant stakeholders.

### 1.2. Embedding Patients’ Perspective into an Operational Framework for Technological Innovation in Rehabilitation Care: The Smart&Touch-ID Approach

Smart&Touch-ID (STID) is a research project funded by the Lombardy Region as part of the HUB Research and Innovation Call (Announcement POR-FESR 2014-2020; Call for strategic research, development and innovation projects aimed at the strengthening of Lombardy ecosystems of research and innovation as hubs with international value; see also Funding section below). The main outcome of the project is the STID model, which structures the design and development of technological solutions for the home-based management of chronic disabilities. The STID model embeds patient evaluations of technology from the very first stages of its development and provides a dedicated infrastructure whereby the different stakeholders can take action as actors of the same ecosystem, not as isolated segments that act sequentially to the same aim.

The vision behind STID consists of harmonizing the health (ID) and well-being (Touch) needs of patients and citizens with the design and development (SMART) actions of Innovators’ eHealth solutions while working on the economic sustainability of the proposed innovation (Governance). This aspect of the STID model provides an answer to the unsolved gap of the patient perspective management in the technology assessment and advancement, by applying a patient-centered and co-design approach, typical of the multidimensional processes [14]. 

The circular model is concretized through the definition of an operational flow that enables interaction between the involved stakeholders and empowers them to carry out their activities in line with their roles and timelines. The flow has been termed ‘circular’ because the realization of activities relevant to the needs of each stakeholder has impacts on the realization of the activities of the other stakeholders in an iterative way. Figure 1 shows a conceptual view of the operational flow. 

An operational flow originates from the launch of a challenge in response to a need. A need models the expression of a demand as perceived by the patient and/or citizen, the domain experts, and the clinicians. A challenge is the process that leads to the identification of technological solutions in response to the detected need that is validated against the ID, Touch, SMART, and Governance aspects.

The evidence about needs that the technological solutions address is collected from different sources (see the upper side of Figure 1). The flow supports two types of challenges to integrate all the identified sources of needs: top-down and bottom-up.

In the top-down challenge, needs are identified by domain experts, on the basis of their in-depth knowledge of the domain, and by clinicians, on the basis of the information they routinely collect from patients during rehabilitation activities. This information is collected in a Register that feeds the modeling of needs in this specific modality.

In the bottom-up challenge, patients and citizens proactively communicate their experiences and attitudes through targeted and anonymous questionnaires (termed “waves”) administered on the website of the project (https://smart-touch-id.com/en/#/home, accessed on 20 May 2023).

Once a challenge is launched, different actors may register for it with different roles. Each role corresponds to different actions that contribute to advancing the challenge. The main roles are Innovators and Citizens: the former propose solutions (prototypes or simple ideas), and the latter manifest interest in the evolution of the challenge and/or declare their interest in testing the solutions.

Innovators who propose ideas are assisted by Experts who help them turn their idea into a prototype through mentorship-dedicated sessions.

All proposed prototypes are validated against SMART, Touch, and ID criteria. If one or more criteria are not met, Experts help Innovators bring their prototypes in line with the established criteria through mentorship. Once prototypes meet all the SMART, Touch, and ID criteria required, they are evaluated for Governance aspects. Finally, prototypes supporting the needs are discussed in terms of their sustainability to identify the most promising ones also from the point of view of patients’ acceptability.

The operational flow has been implemented using PROCS (Process Oriented Development) software, version 1.1.5 [16]. PROCS enables the management and control of complex and dynamic processes. The peculiarity of PROCS is that it allows one to “design” the implementation of each stage of the process and adapt it to the specific needs of the project.

The way STID operates is framed through multidimensional assessment tools, in line with the mainstream methodology for public health interventions. At the same time, its organizational flow takes into account the needs of multiple stakeholders to develop technological solutions that are relevant and applicable to the domain of rehabilitation care, tapping not only the clinical and economic dimensions but also the ones connected to well-being in the real-life experiences of patients [17]. Therefore, STID not only optimizes the multi-stakeholder innovation process in rehabilitation but also enables the construction and stabilization of shared practices for this purpose, activating the process of multidimensional assessment essential to validate the technologies [18]. Further, the definition of the operational flow described above enables the various stakeholders both to tune and perform their activities coherently with their roles and deadlines. This way, it can respond to both public health and patients’ management needs, while facilitating the process of multidimensional assessment to offer innovative and customized solutions for the rehabilitation of chronic disabilities [19].

The designed model takes concrete form through a web portal (https://smart-touch-id.com/en/#/home, accessed on 20 May 2023). People interested in a challenge can subscribe to the website and contribute to it in different ways according to their role (e.g., an innovator can propose an idea or a prototype).

To summarize, challenges operate as “incubators” of technological solutions, allowing the optimization of a suite of solutions that can strengthen community-based rehabilitation of chronic conditions and enhance the competitiveness of the system.

Needs are the core of the model. They can be identified by domain experts and clinicians, but also by patients and citizens. In the present manuscript, the focus is on how patients’ and citizens’ experiences and attitudes on rehabilitation care, detected through “waves”, are embedded into the operational flow of the STID model, providing preliminary evidence for its feasibility under this specific regard.

Materials, methods, and results of the wave of detection on patients’ and citizens’ experiences on rehabilitation are described below, and preliminary results are discussed. 

## 2. Materials and Methods

### 2.1. Participants and Procedure

The data on rehabilitation experiences of patients and citizens were collected via an anonymous online survey generated with the Qualtrics platform using a snowball sampling method but considering the consistent presence of chronic patients equal to 50% of the sample, in accordance with the national data on NCD prevalence (suggesting the impact of at least one NCD in the 50,71% of the population over 25 years old—ISTAT, National Institute of Statistics, 2023 [20]). People could access the questionnaire through an anonymous link on the STID website (https://smart-touch-id.com/#/waves, accessed on 20 May 2023) without having to authenticate. This link was available in the news published on the Regional Open Innovation platform, in the newsletters of Regional and National Clusters, and on the sites of the project partners. People could fill in the survey by accessing it from a computer and mobile devices, such as smartphones and tablets. The participation was anonymous and voluntary, and respondents could abandon completing the questionnaire without consequences. The questionnaire is still accessible through the STID website. The preliminary data here presented were collected from 22 September to 9 November 2022.

The online questionnaire collected demographic information (i.e., age, gender, living condition, education, and working status) and information about the presence of chronic disabilities. Moreover, it collected information about the frequency and perceived proficiency in the use of technologies of all the samples and perceived health-related Quality of Life and rehabilitation experiences of people with chronic disabilities. The following paragraph describes the measures in detail.

Eligible participants were adults over 18 with sufficient Italian language and computer skills to answer the online questionnaire.

The study was conducted following the Declaration of Helsinki and all relevant guidelines and regulations covering respect for the rights and dignity of participants and guaranteeing the anonymity of the data and respecting privacy criteria for respondents.

### 2.2. Measures

#### 2.2.1. Frequency and Perceived Expertise in the Use of Technologies

All participants reported the frequency of use and the perceived expertise of different technological devices (smartphone, mobile phone without touch technology, tablet, computer, smartwatch, voice home assistant). Moreover, they reported the frequency of use and the perceived proficiency in different activities with technologies, namely using the Internet, e-mail, or social networks, e-gaming, reading online magazines, listening to music, or monitoring one’s health with digital devices.

#### 2.2.2. Health-Related Quality of Life and Perceived Health of Participants with Chronic Disabilities

The perceived Quality of Life of respondents with chronic disabilities was measured through the WHODAS 2.0 scale [21]. The World Health Organization (WHO) developed and validated this tool to measure the impact of disability in six domains of functioning. They comprise cognitive activities (understanding and communicating), mobility (moving), self-care (providing personal hygiene, dressing, eating, and standing alone), interpersonal relationships (interacting with other people), daily activities (taking care of the home and family, work and going to school), and participation (taking part in community initiatives, participating in social life and having fun).

For this survey, the 12-item self-report version was used. For each domain, WHODAS 2.0 provides a profile and a summary measure of the level of disability and functioning in daily life applicable to adult and older populations in different cultural contexts. Furthermore, WHODAS 2.0 includes three additional questions that evaluate the frequency and the total or partial impact that the difficulties have on daily life (in the last 30 days). Scoring is on a five-point Likert scale ranging from “no difficulty” to “very difficult.” The scores of all six domains are added. Finally, the total score is converted into a scale from 0 (no disability) to 100 (total disability). High scores, therefore, indicate high levels of disability.

Perceived health was assessed through two questions created ad hoc. Participants had to move a cursor and stop between 0 (the worst level of health) and 10 (the best level), corresponding to how they perceived their health on that day or in the last week.

#### 2.2.3. Rehabilitation Experiences of Participants with Chronic Disabilities

Participants with chronic disabilities answered questions referring to their rehabilitation experiences for their prevalent disability (the one that had the most significant impact on daily life). It was asked if a rehabilitation program had ever been proposed. If so, the regimen (hospitalization, outpatient, or home care), the clinical objective (cognitive rehabilitation, motor rehabilitation, speech therapy, and occupational therapy), the possible use of technological devices, and the nature of such devices (for example, virtual reality headset, tablet, robotic gloves, exoskeleton) were explored in detail.

#### 2.2.4. Basic Attitudes and Beliefs towards Rehabilitation

Basic attitudes and beliefs about rehabilitation were explored among all respondents. The intent was to collect what respondents believed to be true (or false) about rehabilitation. Attitudes and beliefs were explored through 21 statements on which respondents could express their agreement on a five-point Likert scale, from 1 (not at all) to 5 (strongly agree). An example item is “Rehabilitation helps to manage the activities of daily life better.” 

### 2.3. Data Analysis

Data analyses were performed using the IBM SPSS Statistics for Windows, version 26.0 (IBM Corp., Armonk, N.Y., USA) and Jamovi (Version 2.2.5, The Jamovi project, 2021, retrieved from https://www.jamovi.org, accessed on 10 November 2022). All statistical tests were two-tailed, and a *p* ≤ 0.05 was considered statistically significant.

Descriptive statistics were calculated on the sample’s sociodemographic and clinical characteristics.

Mean and standard deviation (SD) were reported for continuous variables and percentages for categorical variables. The items of the scales created ad hoc for the present study underwent a preliminary analysis to check the normal distribution by calculating mean, standard deviation (SD), and indices of skewness and kurtosis. West and colleagues [22] recommend concern if skewness > |2| and kurtosis > |7|.

Mean scores of two original scales (frequency and perceived proficiency in using technologies) were calculated and analyzed. Therefore, their dimensionality was preliminary tested through Exploratory Factor Analysis (EFA). This is the most common approach to scale development and validation [23]. The Kaiser Meyer Olkin (KMO) and Bartlett’s test of sphericity were run to be sure that the correlation matrix could be subjected to analyses. KMO should be > 0.5, and Bartlett’s test of sphericity should be significant. The Maximum Likelihood (ML) estimation method was used. The criterion of eigenvalue greater than one, analysis of the scree plot, and explained variance determined the best-fitting factor solution. In the first step, all items were included. Subsequent factor analyses were conducted stepwise to eliminate items until a stable factor solution was found. Items with a factor loading < 0.32 were excluded. Loadings in the 0.32 range or above are generally considered the cut-off on substantial loadings [24].

Cronbach’s α was calculated to examine the internal consistency of the scales. Cronbach’s α higher than 0.60 was considered acceptable [25].

The association of the frequency and perceived proficiency in the use of technology scores with the relevant sociodemographic variables (i.e., age, gender, and working status) was investigated. Moreover, the differences in dispositions toward rehabilitation between participants who had rehabilitation experience and participants who had not were explored.

The associations between continuous variables (namely, the frequency and perceived proficiency in the use of technologies scores and age) were evaluated by correlations. Following guidelines by Cohen [26], correlations were interpreted as measures of effect size. Correlations were considered weak (|0.10| < r < |0.29|), moderate (|0.30| < r < |0.49|), or strong (|0.50| < r < |1|).

The associations between continuous variables (namely, the frequency and perceived proficiency in the use of technologies scores and the dispositions towards rehabilitation scores) and categorical variables (i.e., gender, working status, and rehabilitation experience) were evaluated by *t*-tests. Assumption checks were performed before each *t*-test by evaluating skewness and kurtosis to check the normal distribution of the variables and Levene’s test to check the variances’ homogeneity. Based on the assumption checks results, *t*-tests were performed using Welch’s Test (normal distribution and unequal variances), Student’s Test (normal distribution and equal variances), or the non-parametric Mann–Whitney U Test.

A Kruskal–Wallis non-parametric one-way ANOVA was performed to test for differences in the health-related Quality of Life reported by participants with different clinical conditions.

## 3. Results

### 3.1. Study Participants

There were 95 participants. The distribution by age group is relatively homogeneous, ranging from 25 to 84 years old, the most represented group being between 65 and 74 years old (24% of the sample). Women are 60% of the sample. Most of the sample (78%) lives with someone else, in most cases (63%) the spouse or partner and children (25%). The sample is quite heterogeneous by educational level. The most represented educational qualification is the high school diploma (26%). Half of the sample is retired (55%). 

More than half of the participants (53 people, 56% of the sample, resulting in line with the expected 51% suggested by the Italian NCD prevalence reported by ISTAT, 2023) declared suffering from at least one chronic disability (nine declared at least one comorbidity). Considering, in the case of comorbidity, only the disability that participants considered prevalent (i.e., the one that most impacts daily activities), the disabilities more frequent are neurological (22 respondents, 24% of the sample) and osteoarticular (22 respondents, 24% of the sample). Cardiovascular and chronic pulmonary disabilities (about 8% and 10% of the sample) are less represented.

Table 1 provides a detailed description of the sample characteristics.

### 3.2. Frequency and Perceived Proficiency in the Use of Technologies

Seventy-seven percent of the participants stated that they used a smartphone daily in the last year. The use of mobile phones without touch technology is infrequent: 68% of participants have never used it in the last year. Tablet usage is quite rare: 51% of the sample has never used it in the last year. The frequency of computer use is more distributed. Most of the sample has never used a digital watch that detects health data (76%) or a vocal home assistant (77%); 65% of the sample in the last year used the internet daily; on the other hand, a considerable percentage of the sample (22%) had never performed it. The usage of social networks split participants between those who use them daily (46%) and those who never used them (43%). Most of the sample (66%) did not use e-games. Reading online material and listening to digital music is an evenly distributed activity; 55% of the sample used e-mail daily in the last year; on the other hand, a considerable percentage of the sample (32%) never used it; 54% of the sample in the last year never monitored their health with digital devices.

The items on the scale “use frequency of technologies” met the criteria to assume their normality (i.e., skewness < |2| and kurtosis < |7|). The Bartlett’s sphericity test [χ^2^(78) = 707, *p* < 0.001] and the KMO (equal to 0.860) ensured that the correlation matrix could be subjected to factor analysis. The analysis indicated that a single-factor solution was the most appropriate. No item displays a loading lower than 0.32 except one (use of the mobile phone without touch technology, see Table 2). Therefore, the initial pool of thirteen items was reduced to twelve. The total variance explained by the factor extracted was 45.1%. The reliability analysis showed that the scale has optimal internal consistency (Cronbach’s alpha = 0.903).

A total “use frequency of technologies” score was calculated based on the mean of the final twelve items; this new variable showed a normal distribution. The use frequency of technologies correlated significantly and negatively with age (r = −0.571; *p* < 0.001). The effect size was large. The older the age, the lower the frequency of use. There was no statistically significant difference in the use frequency between men and women [Student’s t(91) = −0.487; *p* = 0.628]. There was a statistically significant difference in the use frequency between workers and retirees [Welch’s t(88) = 6.883; *p* < 0.001]. Working people used technologies more frequently (mean = 3.50, range 1–5) than retirees (mean = 2.25). This result overlapped with the result for age (retirees are usually also the oldest). Table 3 shows the data on the use frequency of technologies of all the respondents, the under 65 and over 65 groups.

Regarding the ratings on one’s own expertise, the majority of participants felt proficient (somewhat or very much) in using a smartphone (64%) and a mobile phone without touch technology (70%). Data about tablet and computer use were more distributed. The perceived proficiency in using smartwatches and vocal home assistants reflects the data on their frequency of use: the majority of the sample does not feel proficient in using these devices (40% and 51%, respectively). Most of the sample (44%) felt expert in internet use. Regarding the navigation of social networks and reading online material, the sample is divided between those who feel very or quite proficient and those who do not. Most of the sample (42%) declared not feeling proficient in e-gaming; on the contrary, they felt very proficient (44%) in using e-mail. Regarding listening to music on digital devices, 35% of the sample felt very proficient, compared with a similar percentage (28%) of respondents who did not feel proficient. The perceived proficiency in using digital health monitoring devices was quite distributed, with a prevalence of respondents who did not feel proficient (34%).

The items of proficiency in the technologies use scale met the criteria to assume their normality (i.e., skewness < |2| and kurtosis < |7|). The Bartlett’s sphericity test [χ^2^(78) = 1270, *p* < 0.001] and the KMO (equal to 0.925) ensured that the correlation matrix could be subjected to factor analysis. The analysis indicated that a single-factor solution was the most appropriate. No item displays a loading lower than 0.32. In this case, the saturation of the item “use of the mobile phone without touch technology” was significant, albeit lower than the other items’ saturations (Table 4). However, this item was eliminated for consistency with the analyses made on the preceding scale. Therefore, the initial pool of thirteen items was reduced to twelve. The total variance explained by the factor extracted was 70.1%. The reliability analysis showed that the scale has optimal internal consistency (Cronbach’s alpha = 0.965).

A total “proficiency in the technologies use” score was calculated based on the mean of the final twelve items; this new variable showed a normal distribution. Perceived proficiency in the use of technologies correlated significantly and negatively with age (r = −0.636; *p* < 0.001). The effect size was large. The older the age, the lower the perceived proficiency in using technologies. There was no statistically significant difference in perceived proficiency between men and women [Student’s t(91) = −0.550; *p* = 0.584]. There was a statistically significant difference in perceived proficiency between workers and retirees [Welch’s t(88) = 7.007; *p* < 0.001]. Workers felt more proficient (mean = 3.96, range 1–5) than retirees (mean = 2.41). As already observed for the frequency of use of technologies, this result overlaps with the result relating to age (retirees were also the oldest). Frequency of use and perceived competence correlated significantly and positively (r = 0.846; *p* < 0.001). The effect size was large. Table 5 shows the data on the proficiency in the technologies used by all the respondents, the under 65 and over 65 groups.

### 3.3. Participants with Chronic Disabilities: Clinical Conditions and Perceived Quality of Life

Participants with chronic disabilities reported a disability level equal to 27.6 (SD = 23.3), indicating a perceived disability of mild to moderate level. The health perceived on the day they completed the questionnaire (mean = 6.72; SD = 1.86; range 1–10) and in the last week (mean = 6.42; SD = 1.98) confirmed this result. The responses to the WHODAS 2.0 and the questions about perceived health had a normal distribution. Nevertheless, the number of participants for each clinical condition was unequal and in the case of cardiovascular and pulmonary disability was low. Therefore, a Kruskal–Wallis non-parametric one-way ANOVA was performed to test for differences in the health-related quality of life reported by participants with different clinical conditions. Results suggested that the WHODAS 2.0 scores and the declared perceived health did not differ significantly in the function of the different clinical conditions (*p* = 0.857; *p* = 0.504; *p* = 0.488, respectively).

### 3.4. Participants with Chronic Disabilities: The Rehabilitation Experience

First, we explored whether participants with chronic disabilities had ever been offered rehabilitation. Approximately 42% of participants with chronic disabilities (22 out of 53 people) had never been advised by a clinician to participate in a rehabilitation program. Instead, 31 respondents in that group (58%) were prescribed it, and only one decided not to follow it.

Among the 30 respondents with experience of at least one rehabilitation process, half had completed the whole treatment cycle, while the other half had ongoing experiences.

Taking into account all the rehabilitation experiences reported by the participants in the various regimes, rehabilitation occurred less frequently during hospitalization (7 cases out of 30) and more often on an outpatient basis at a clinical center (22 respondents) or at home (10 respondents).

Considering that participants could have undergone more than one rehabilitation cycle under different regimens and with different clinical objectives (for example, motor rehabilitation, cognitive rehabilitation, or speech therapy), they were asked to respond by thinking of the last rehabilitation cycle carried out in a specific regimen. Of the total rehabilitation experiences reported by the 30 respondents, the type of rehabilitation performed was mainly motor (a total of 38 experiences were reported between inpatient, outpatient, and home care). Less frequent were the experiences of cognitive rehabilitation (seven cases), speech therapy (three cases), or occupational therapy (one case).

Regardless of the type of rehabilitation, it emerges that the use of technologies in the rehabilitation context (that is, asking the patient if he had personally used—alone or with support from the therapist—technologies specifically aimed at carrying out the rehabilitation activity) was limited and referred to specific rehabilitation contexts. Among those who reported experiences in an outpatient setting, about a third (7 cases out of 22) used technological devices. Much less frequent, in the experience of the group described, was the use of technological devices for rehabilitation at home (2 out of 10). On the other hand, no one reported having undergone a rehabilitation cycle with technological devices in hospitalization.

Technologies have been an enabling factor mainly for motor rehabilitation (wearable sensors, four cases; technological platform/treadmill, four cases; virtual reality headset/headset, robotic gloves, and exoskeleton, each in a single case). On the other hand, the computer/tablet was more present in cognitive rehabilitation experiences (in five out of six cases, except for one case in which it was used for motor rehabilitation).

### 3.5. Basic Attitudes and Beliefs towards Rehabilitation

Attitudes and beliefs about rehabilitation were explored among all respondents (*n* = 95). The first result is that, regardless of age, health conditions, and rehabilitation experience, participants believe that rehabilitation is a “useful”, “important”, and “necessary for health” activity. For these three statements, the variable had a strong negative asymmetry; in other words, most respondents agreed very much (ceiling effect). the position of the respondents was more nuanced on the accessibility of rehabilitation (“It is accessible to anyone who wants to carry it out”, “It is affordable”, “It does not take too long to go where they do it”; see Table 6).

Considering the comparison between participants with and without rehabilitation experience, the following additional findings are also noteworthy (Table 6). First, beliefs about the positive impact of rehabilitation in managing activities of daily living were significantly more pronounced among those who had experienced it (mean = 4.37) than among those who had never performed it (mean = 3.74; *p* = 0.053). Second, the valuing of rehabilitation as a necessary constituent of one’s well-being was significantly more present among respondents who had experienced it (mean = 4.57) than among respondents who had not (mean = 3.87; *p* < 0.01). Third, the belief that rehabilitation improves mood was significantly more prevalent among those who had experienced it (mean = 4.07) than those who had not (mean = 3.52; *p* < 0.05). Fourth, the perception of the fatigue value of doing rehabilitation was significantly more marked among those who had experienced it (mean = 4.57) than those who had not (mean = 3.87; *p* < 0.05). Finally, the perception of the importance of rehabilitation was significantly clearer among those who had experienced it (mean = 4.70) than in the other group (mean = 4.09; *p* < 0.05).

## 4. Discussion

Though preliminary and based on data collected through random waves of detection, our results are in line with the core message by Cieza and colleagues [4]: only 43% of respondents with NCDs were prescribed at least a rehabilitation treatment since diagnosis. Moreover, among respondents with NCDs, only a few participants (17%) were delivered tech-enhanced rehabilitation treatments, mainly as outpatients in hospital settings. Further, considering that participants with NCDs perceived the impact of disability on their everyday routines mild to moderate, it can be assumed that these people were in the early stages of their disease, that is, in those stages of the condition that would benefit more from access to rehabilitation pathways [27]. Besides the patient experiences, it is also interesting that both citizens (without a direct personal experience with NCDs) and patients believe that rehabilitation is a service for the few: it is not considered a service that one can have easy access to; the sites where it is delivered are not seen as easy to reach. Beliefs indeed have an impact on behaviors, since they regard how one makes sense of one’s own experiences and represent what one can expect from one’s own concrete actions. If a mainstream belief in rehabilitation care is that there is an issue regarding its equitable access, there is virtually no reason for patients and citizens to take steps to reach this aim. Beliefs are at the core of the networks of knowledge and meanings that define culture not as a stable and mono-dimensional entity, but as a constellation of learned routines of thinking and interacting with other people [28]. Although these networks are not often unitary even within the same group of people, this is not the case in the specific domain of rehabilitation care: both from the academic research standpoint and from the patients’ and citizens’ standpoint, there is a substantial alignment on the conviction that equitable access to patients in need is, so far, not available.

Although a growing body of evidence shows that a technological boost is transforming rehabilitation by modeling continuity of care pathways from hospital to home, still the main question for the HTA evaluation approach is: does it work? To answer the question, besides validation of clinical benefits, key domains in this multidimensional approach are the usability and the acceptability of a technological solution. It is known that lack of usability provides a substantial barrier to the adoption of new health technologies [29]. Considering the trade-off between perceived advantages and effort in the experience with technology, acceptability evaluations regard how beliefs on perceived ease of use and perceived usefulness impact the intention to use a specific technology (Technology Acceptance Model, TAM, [30]). Further, Tsertsidis and colleagues [31] have pointed out that demographic characteristics and technological expertise can also influence the user acceptance of technology. Results from the feasibility study conducted show that perceived technological expertise and the use of digital devices are negatively correlated with age, with significant differences between over 65 and under 65 participants: the older the age, the lower the perceived proficiency in using technologies and the use of ICT technologies in everyday routines. As obvious as this result can be, it highlights that even taking into account the upcoming increase in technological expertise due to the aging of current younger (under 65) generations, a gap is still in place considering the aging and the increased life expectancy of the global population. This gap, in turn, calls for the demand on health services to deal with disabling outcomes, and for policymakers to anticipate these changes [3].

To this aim, in the STID perspective, when it comes to patients’ involvement in evaluating the usability and acceptability of technologies that enable rehabilitation treatments, the starting point is not the technological devices that the patient can “use” and can “accept”, but the patient’s experience that such technologies enable. That is, evaluations on the usability and acceptability of a technological solution are an integral part of the design and development process of a technology, in a multi-stakeholder (patients, industries, clinicians, HTA experts) co-design methodological and organizational framework. The design and development of a technology that includes patients from the very first steps of its development is indeed at the core of the User-Centered Design (UCD) methodological approach [32]. According to this methodological option, the dovetailing of users’ intentions (i.e., what the user wants to do with technology) with the design of the features and functions of a specific interactive system is the outcome of an iterative process where consecutive cycles of design-evaluation-redesign of technology are carried out to optimize the experience of the user [32,33]. Along this line, in STID, technologies are viewed as interfaces enabling rehabilitation experiences whose constraints (regarding design interaction, narrative options, graphical interface, data model, etc.) pertain, in an iterative fashion, both to the actors of the rehabilitation experience (patients and clinicians) and to actors of the technological and manufacturer industry. This way, the STID flow incorporates end-user evaluations on the perceived ease of use and acceptability of technology into the design and development process of the technology itself. Moreover, operationally, through the dynamic interplay between Challenges (as technological incubators) and Waves (as systematic and continuous detection of patients’ and citizens’ experiences on rehabilitation), the STID platform provides the organizational structure for this multi-stakeholder co-design.

The vision behind STID whereby technologies for rehabilitation are not devices that the patient uses but consist in constraints that shape the precise experience that they enable implies that the set-up of a technology for rehabilitation consists in the sense-making process that, though a technology, one can cue and activate. That is, it assumes that technologies do not add up to their context of clinical use, but are an integral part of it [34,35,36]. Drawing on this assumption, a substantial section of the questionnaire administered to people with direct or indirect experience of NCDs was devoted to the detection of their attitudes and beliefs on rehabilitation. The results detailed above show that, first of all, both patients and citizens value rehabilitation as useful and necessary for one’s health. Further, although shared also among people who had no personal experience of rehabilitation, it is noteworthy that people with NCDs who reported at least one rehabilitation experience significantly highlighted more than others the belief that rehabilitation has a substantial positive impact on the management of daily care routines, on their well-being, and on their mood. In other words, the health outcomes expected from rehabilitation care are not separate, in their point of view, from well-being and quality of life outcomes. The perceived benefits and expectations on rehabilitation, therefore, go beyond health outcomes, turning rehabilitation into a holistic experience, which is able to influence also the affective and cognitive dimensions of managing life with a chronic condition. 

These results on the patient’s perspective on rehabilitation fit into the STID flow for the introduction of new technologies since they provide further dimensions besides health outcomes (as a basic requirement) to be integrated into their design and development. Specifically, the integration of quality of life and well-being outcomes. This theme may not be acknowledged by Innovators (i.e., industries), resulting in the design and development of technological solutions that, after a considerable effort in prototype development, run the risk of being evaluated as only partially suitable to meet patients’ needs. The assumption, on the Innovators’ side, may be that technology can be pushed to optimal development without the involvement of patients and citizens since its early prototypical design. In this organizational setup, with pilot studies with end users evaluating existing and quite defined prototypes, hassles connected with multidisciplinary design and development workgroups are minimized. Nonetheless, segmenting the development trajectory into defined parcels increases the risk of having to reorganize the production line, especially if the technological prototype tested in real-life settings fails to address patients’ needs with negative impacts on adherence to treatment, and efficacy. Differently, the early effort connected with a development pipeline of multistakeholder co-design and evaluations provides a more efficient option. 

Along this line, a further result is that at least two different networks of knowledge and belief are at stake in the ecosystem of technological innovation for rehabilitation care: one shared by Innovators, and one shared by patients, that may not overlap. The STID model acknowledges that such networks of knowledge may—and often are not—consistent. Therefore it provides an organizational flow that not only allows Innovators and patients to have their respective knowledge accessible in one’s own cognitive repertoire but also makes it applicable to the specific domain of rehabilitation care. That is, the STID flow manages to turn the perceived value of respective beliefs (of Innovators and patients) into operational pathways that become applicable to the task at hand. This way, it provides the operational framework to build a community of practices able to transform the culture of rehabilitation care including the patient’s perspectives since the early process of technological innovation. 

## 5. Limitations and Conclusions

The main limitation of the present study is related to the uncertainty of the real possibility to implement innovation in rehabilitation care processes, not depending directly on the STID model, but on the decision-making process taking place at a regulatory level. 

To date, the public health supply of rehabilitation treatments for people with Non-Communicable diseases is not able to cover the population’s needs. This misalignment between supply and demand depends, on the one hand, on the inability to guarantee adequate services to all patients in need and, on the other hand, on the difficulty of fully exploiting the potential of technology to innovate rehabilitation. Low capacity and delayed access are also due to outdated and poorly updated reimbursement systems and tariffs not being in line with the current technological opportunities. This aspect may also discourage Innovators since they could not find adequate pricing in relation to their purpose to explore new approaches and/or solutions.

The STID model, combining patients’ and citizens’ perspectives with the Governance dimension, would support the public service to solve this specific problem, not only developing the most promising solutions for the market but also providing clear indications of how they can be reimbursed by the healthcare service. 

Indeed, in the STID model, the patient-citizen perspectives on innovative technologies for rehabilitation are integrated into an operational flow for the creation of innovative, needs- and territory-connected, personalized, and sustainable rehabilitation systems.

In STID, the detection of patient health and well-being experiences on rehabilitation works as an integral dimension in the design and development of technologies, providing both in itinere and endpoint evaluations. Since the STID flow also takes into account the needs of the other relevant stakeholders, it taps several dimensions—i.e., the clinical, the economic, and the well-being ones—able to activate the process of multidimensional assessment essential to validate the technologies in line with public health demands. By optimizing the multi-stakeholder innovation process in rehabilitation, STID also enables the construction and stabilization of shared practices for the purpose of offering innovative and customized solutions for the rehabilitation of chronic disabilities.

The main strength of the STID model is that it allows the management of patients’ rehabilitation perspectives and evaluations on health and well-being in rehabilitation while enabling their turning into operational practices into a multi-stakeholder co-design framework. This way, the ecosystem of needs and opportunities of all stakeholders in tech-enhanced rehabilitation care is nurtured by distributed shared networks of knowledge, to tune the respective actions and stabilize a participating culture of multidimensional evaluation for technological innovation and advancement. Although preliminary evidence supports this claim, the outcomes of these development and co-creation frameworks require that in the regulatory domain, active pathways for the definition of rates that include specific technologies, or proactive innovative mechanisms for reimbursement are set up and defined, to let such process and organizational innovations transform the public health assets to concretely meet the growing need of rehabilitation for people with NCDs.

## Figures and Tables

**Figure 1 healthcare-11-01604-f001:**
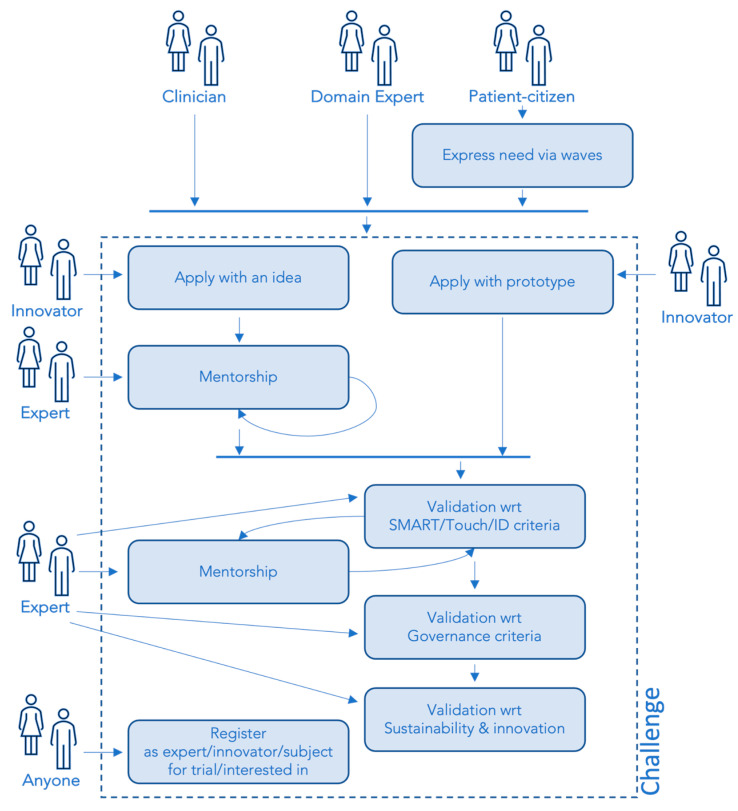
The operational flow of the STID model.

**Table 1 healthcare-11-01604-t001:** Sociodemographic and clinical characteristics of the sample (*n* = 95).

		Frequency	Percentage
Age	24 or less	4	4.2
	25–34	9	9.5
	35–44	11	11.6
	45–54	10	10.5
	55–64	16	16.8
	65–74	23	24.2
	75–84	16	16.8
	85 or more	6	6.3
Gender	Man	36	37.9
	Woman	57	60.0
	Other/Prefer not to state	2	2.1
Living condition (more than one answer possible)	Alone	22	23.2
	Partner	60	63.2
	Children	24	25.3
	Caregiver	1	1.1
	A family member (other than a partner and children)	6	6.3
	Someone else	1	1.1
Education	Primary school diploma	12	12.6
	Middle School diploma	22	23.2
	High school diploma	25	26.3
	Degree	23	24.2
	Postgraduate training	13	13.7
Working status	Paid worker	38	40.0
	Retired	52	54.7
	No paid work	5	5.3
Prevalent chronic disability	Neurological	22	24.2
	Cardiovascular	3	8.4
	Pulmonary	6	9.5
	Osteoarticular	22	24.2
	No disability	42	44.2

**Table 2 healthcare-11-01604-t002:** Factor loadings resulting from the EFA about the “use frequency of technologies” scale.

Items of the “Use Frequency of Technologies” Scale	Factor Loadings
E-mail	0.931
Internet	0.878
Computer	0.842
Social network	0.785
Online reading	0.745
Smartphone	0.730
Digital Music	0.685
Tablet	0.528
E-gaming	0.468
Monitoring health with a device	0.390
Smartwatch	0.363
Voice home assistant	0.329
A mobile phone without touch technology	0.208

**Table 3 healthcare-11-01604-t003:** Frequency distribution of technologies use (*n* = 95).

Use of …	Age	Never	Once a Month or More Rarely	Sometimes a Month	A Few Times a Week	Everyday
Smartphone	All sample	16 (16.8%)	2 (2.1%)	1 (1.1%)	3 (3.2%)	73 (76.8%)
	Under 65	4 (8.0%)	\	\	1 (2.0%)	45 (90.0%)
	Over 65	12 (26.7%)	2 (4.4%)	1 (2.2%)	2 (4.4%)	28 (62.2%)
A mobile phone without touch technology	All sample	65 (68.4%)	9 (9.5%)	1 (1.1%)	2 (2.1%)	18 (18.9%)
	Under 65	40 (80.0%)	2 (4.0%)	1 (2.0%)	\	7 (14.0%)
	Over 65	25 (55.6%)	7 (15.6%)	\	2 (4.4%)	11 (24.4%)
Tablet	All sample	50 (52.6%)	10 (10.5%)	13 (13.7%)	11 (11.6%)	11 (11.6%)
	Under 65	17 (34.0%)	8 (16.0%)	9 (18.0%)	7 (14.0%)	9 (18.0%)
	Over 65	33 (73.3%)	2 (4.4%)	4 (8.9%)	4 (8.9%)	2 (4.4%)
Computer	All sample	31 (32.6%)	7 (7.4%)	5 (5.3%)	9 (9.5%)	43 (45.3%)
	Under 65	6 (12.0%)	4 (8.0%)	1 (2.0%)	6 (12.0%)	33 (66.0%)
	Over 65	25 (55.6%)	3 (6.7%)	4 (8.9%)	3 (6.7%)	10 (22.2%)
Smartwatch	All sample	72 (75.8%)	3 (3.2%)	1 (1.1%)	5 (5.3%)	14 (14.7%)
	Under 65	32 (64.0%)	3 (6.0%)	1 (2.0%)	5 (10.0%)	9 (18.0%)
	Over 65	40 (89.0%)	\	\	\	5 (11.1%)
Voice home assistant	All sample	73 (76.8%)	5 (5.3%)	2 (2.1%)	3 (3.2%)	12 (12.6%)
	Under 65	33 (66.0%)	5 (10.0%)	\	2 (4.0%)	10 (20.0%)
	Over 65	40 (88.9%)	\	2 (4.4%)	1 (2.2%)	2 (4.4%)
Internet	All sample	21 (22.1%)	4 (4.2%)	3 (3.2%)	5 (5.3%)	62 (65.3%)
	Under 65	2 (4.0%)	2 (4.0%)	2 (4.0%)	1 (2.0%)	43 (86.0%)
	Over 65	19 (42.2%)	2 (4.4%)	1 (2.2%)	4 (8.9%)	19 (42.2%)
Social network	All sample	41 (43.2%)	2 (2.1%)	1 (1.1%)	7 (7.4%)	44 (46.3%)
	Under 65	11 (22.0%)	1 (2.0%)	1 (2.0%)	5 (10.0%)	32 (64.0%)
	Over 65	30 (66.7%)	1 (2.2%)	\	2 (4.4%)	12 (26.7%)
E-gaming	All sample	63 (66.3%)	4 (4.2%)	8 (8.4%)	8 (8.4%)	12 (12.6%)
	Under 65	24 (48.0%)	3 (6.0%)	7 (14.0%)	7 (14.0%)	9 (18.0%)
	Over 65	39 (86.7%)	1 (2.2%)	1 (2.2%)	1 (2.2%)	3 (6.7%)
Online reading	All sample	30 (31.6%)	6 (6.3%)	11 (11.6%)	21 (22.1%)	27 (28.4%)
	Under 65	5 (10.0%)	5 (10.0%)	9 (18.0%)	12 (24.0%)	19 (38.0%)
	Over 65	25 (55.6%)	1 (2.2%)	2 (4.4%)	9 (20.0%)	8 (17.8%)
E-mail	All sample	30 (31.6%)	3 (3.2%)	2 (2.1%)	8 (8.4%)	52 (54.7%)
	Under 65	4 (8.0%)	3 (6.0%)	\	6 (12.0%)	37 (74.0%)
	Over 65	26 (57.8%)	\	2 (4.4%)	2 (4.4%)	15 (33.3%)
Digital Music	All sample	31 (32.6%)	10 (10.5%)	7 (7.4%)	23 (24.2%)	24 (25.3%)
	Under 65	7 (14.0%)	6 (12.0%)	3 (6.0%)	14 (28.0%)	20 (40.0%)
	Over 65	24 (53.3%)	4 (8.9%)	4 (8.9%)	9 (20.0%)	4 (8.9%)
Monitoring health with a device	All sample	51 (53.7%)	8 (8.4%)	10 (10.5%)	12 (12.6%)	14 (14.7%)
	Under 65	23 (46.0%)	3 (6.0%)	8 (16.0%)	9 (18.0%)	7 (14.0%)
	Over 65	28 (62.2%)	5 (11.1%)	2 (4.4%)	3 (6.7%)	7 (15.6%)

**Table 4 healthcare-11-01604-t004:** Factor loadings resulting from the EFA about the “proficiency in the technologies use” scale.

Items of the “Proficiency in the Technologies Use” Scale	Factor Loadings
Internet	0.923
Online reading	0.909
Computer	0.908
Smartphone	0.899
Tablet	0.892
E-mail	0.887
Social network	0.852
Digital Music	0.844
E-gaming	0.757
Monitoring health with a device	0.740
Smartwatch	0.722
Voice home assistant	0.659
A mobile phone without touch technology	0.430

**Table 5 healthcare-11-01604-t005:** Frequency distribution of proficiency in the technologies use (*n* = 95).

Use of …	Age	Not at All	A Little	Neither a Little nor Enough	Enough	Very
Smartphone	All sample	12 (12.6%)	11 (11.6%)	12 (12.6%)	31 (32.6%)	29 (30.5%)
	Under 65	1 (2.0%)	3 (6.0%)	3 (6.0%)	20 (40.0%)	23 (46.0%)
	Over 65	11 (24.4%)	8 (17.8%)	9 (20.0%)	11 (24.4%)	6 (13.3%)
A mobile phone without touch technology	All sample	4 (4.2%)	11 (11.6%)	14 (14.7%)	35 (36.8%)	31 (32.6%)
	Under 65	3 (6.0%)	4 (8.0%)	5 (10.0%)	17 (34.0%)	21 (42.0%)
	Over 65	1 (2.2%)	7 (15.6%)	9 (20.0%)	18 (40%)	10 (22.2%)
Tablet	All sample	31 (32.6%)	6 (6.3%)	13 (13.7%)	23 (24.2%)	22 (23.2%)
	Under 65	5 (10.0%)	1 (2.0%)	10 (20.0%)	16 (32.0%)	18 (36.0%)
	Over 65	26 (57.8%)	5 (11.1%)	3 (6.7%)	7 (15.6%)	4 (8.9%)
Computer	All sample	25 (26.3%)	8 (8.4%)	8 (8.4%)	26 (27.4%)	28 (29.5%)
	Under 65	3 (6.0%)	4 (8.0%)	5 (10.0%)	16 (32.0%)	22 (44.0%)
	Over 65	22 (48.9%)	4 (8.9%)	3 (6.7%)	10 (22.2%)	6 (13.3%)
Smartwatch	All sample	38 (40%)	13 (13.7%)	10 (10.5%)	20 (21.1%)	14 (14.7%)
	Under 65	8 (16.0%)	8 (16.0%)	8 (16.0%)	16 (32.0%)	10 (20.0%)
	Over 65	30 (66.7%)	5 (11.1%)	2 (4.4%)	4 (8.9%)	4 (8.9%)
Voice home assistant	All sample	48 (50.5%)	12 (12.6%)	9 (9.5%)	13 (13.7%)	13 (13.7%)
	Under 65	17 (34.0%)	7 (14.0%)	5 (10.0%)	8 (16.0%)	13 (26.0%)
	Over 65	31 (68.9%)	5 (11.1%)	4 (8.9%)	5 (11.1%)	\
Internet	All sample	16 (16.8%)	10 (10.5%)	6 (6.3%)	21 (22.1%)	42 (44.2%)
	Under 65	1 (2.0%)	1 (2.0%)	3 (6.0%)	14 (28.0%)	31 (62.0%)
	Over 65	15 (33.3%)	9 (20%)	3 (6.7%)	7 (15.6%)	11 (24.4%)
Social network	All sample	30 (31.6%)	10 (10.5%)	11 (11.6%)	20 (21.1%)	24 (25.3%)
	Under 65	6 (12.0%)	5 (10.0%)	5 (10.0%)	16 (32.0%)	18 (36.0%)
	Over 65	24 (53.3%)	5 (11.1%)	6 (13.3%)	4 (8.9%)	6 (13.3%)
E-gaming	All sample	40 (42.1%)	9 (9.5%)	14 (14.7%)	15 (15.8%)	17 (17.9%)
	Under 65	7 (14.0%)	6 (12.0%)	9 (18.0%)	13 (26.0%)	15 (30.0%)
	Over 65	33 (73.3%)	3 (6.7%)	5 (11.1%)	2 (4.4%)	2 (4.4%)
Online reading	All sample	21 (22.1%)	6 (6.3%)	9 (9.5%)	26 (27.4%)	33 (34.7%)
	Under 65	1 (2.0%)	2 (4.0%)	5 (10.0%)	16 (32.0%)	26 (52.0%)
	Over 65	20 (44.4%)	4 (8.9%)	4 (8.9%)	10 (22.2%)	7 (15.6%)
E-mail	All sample	28 (29.5%)	4 (4.2%)	2 (2.1%)	19 (20.0%)	42 (44.2%)
	Under 65	3 (6.0%)	2 (4.0%)	1 (2.0%)	11 (22.0%)	33 (66.0%)
	Over 65	25 (55.6%)	2 (4.4%)	1 (2.2%)	8 (17.8%)	9 (20.0%)
Digital Music	All sample	27 (28.4%)	8 (8.4%)	6 (6.3%)	21 (22.1%)	33 (34.7%)
	Under 65	3 (6.0%)	5 (10.0%)	4 (8.0%)	9 (18.0%)	29 (58.0%)
	Over 65	24 (53.3%)	3 (6.7%)	2 (4.4%)	12 (26.7%)	4 (8.9%)
Monitoring health with a device	All sample	32 (33.7%)	11 (11.6%)	14 (14.7%)	20 (21.1%)	18 (18.9%)
	Under 65	8 (16.0%)	7 (14.0%)	7 (14.0%)	13 (26.0%)	15 (30.0%)
	Over 65	24 (53.3%)	4 (8.9%)	7 (15.6%)	7 (15.6%)	3 (6.7%)

**Table 6 healthcare-11-01604-t006:** Descriptive statistics of dispositions about rehabilitation (*n* = 95) and group differences between patients with rehabilitation experience vs. citizens without rehabilitation experience.

Rehabilitation …	Mean	SD	Asym.	Kurt.	Statistic	df	*p*
is useful	4.58	0.88	−2.67	7.49	(M) 1.65	/	0.102
is an engaging activity	3.98	1.02	−0.88	0.23	(S) −0.97	51	0.338
helps to manage the activities of daily life better	4.18	1.00	−1.35	1.66	(W) −2.00	37	0.053
is necessary for health	4.58	0.77	−2.59	8.68	(M) 0.76	/	0.457
makes feel capable of doing things	4.13	1.03	−1.29	1.32	(S) −1.48	51	0.146
helps live better	4.43	0.84	−1.85	4.41	(S) −0.74	51	0.461
is accessible to anyone who wants to do it	2.94	1.38	0.12	−1.25	(S) −0.05	51	0.959
helps manage the disability better	4.22	0.91	−1.35	2.14	(S) −1.62	51	0.111
it is important	4.52	0.89	−2.44	6.51	(M) 2.15	/	0.032
helps to accept one’s illness	3.66	1.22	−0.47	−0.92	(S) −0.00	51	0.997
is necessary for the well-being	4.41	0.91	−1.98	4.49	(S) −2.71	51	0.009
makes feel the protagonist of the treatment	4.05	0.98	−0.96	0.64	(S) −1.33	51	0.191
is not boring	3.58	1.25	−0.40	−0.85	(S) −0.49	50	0.624
helps understand one’s illness	3.53	1.08	−0.52	−0.41	(W) −1.02	51	0.311
it is worth the effort	4.34	0.96	−1.79	3.33	(W) −2.43	34	0.020
helps feel better with others	3.76	1.06	−0.70	−0.11	(S) −1.07	51	0.288
helps to make you independent in everyday life	4.19	1.07	−1.36	1.19	(S) −1.33	51	0.191
improves mood	3.89	0.98	−0.63	0.08	(S) −2.08	51	0.043
it does not take too long to get to where they make it	3.24	1.31	−0.30	−0.90	(W) −0.16	51	0.876
is affordable	2.97	1.24	0.13	−0.85	(W) −0.36	51	0.719
is always different	2.95	0.96	−0.04	0.00	(S) 0.71	51	0.483

Note. The scale ranges from 1 to 5. Asym = Asymmetry (Standard Error of Asymmetry = 0.249). Kurt = Kurtosis (Standard Error of Kurtosis = 0.493). M = standardized Mann–Whitney’s U; S = Student’s t; W = Welch’s t.

## Data Availability

Data can be obtained upon reasonable request from the corresponding author.

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
