# Peer review of "Embedding the Patient-Citizen Perspective into an Operational Framework for the Development and the Introduction of New Technologies in Rehabilitation Care: The Smart&Touch-ID Model"

_healthcare, 2023, doi:10.3390/healthcare11111604_

Round 1

Reviewer 1 Report

The manuscript is well-written in clear, concise language in addition to logical and easy-to-follow and understand way. The flow of the write-up is commendable.

The paper has a clear objective on the application of the Smart&Touch-ID Model as a framework for developing and introducing new technologies in rehabilitation care. The introduction of the paper contains adequate background information to help readers understand and follow the argument in a well-justified study paper. However, what I did not find and hope I’m not missing are clear examples of rehabilitation care and what some examples might entail. When we talk about morbidities requiring rehabilitation care one tends to think more of morbidities associated with movement and other chronic disabilities such as lymphedema, strokes, and chronic would requiring long-term wound care, for which such models and technologies could be equally essential. Regarding the technologies investigated, though various technologies and platforms were investigated those with video capability for conferencing or 1:1 connectivity should have been specifically mentioned and described in some more detail while emphasizing their utility and potential for use particularly in these times as potential replacements for patient-doctor consultations and follow-ups. In rehabilitative care, peer-group meetings involving sharing experiences, expectations, and learnings from experts have played a great role in the success of such strategies in rehabilitation care. This is an aspect that I would have thought was potentially explored. Lessons from these aspects would probably have enriched the framework.

I think the methodology applied was robust, and the statistical analysis was detailed and led to very pertinent conclusions and recommendations on the design of an operational framework for the development and introduction of new technologies in rehabilitation care.

I think this paper particularly in this era where virtual meetings and online engagements and platforms to facilitate these processes are gaining prominence this manuscript will hold a special appeal to the global health constituency for its scientific practical value. Very little editorial work is required in the paper.

I recommended the paper for publication.  

Author Response

We thank you very much for appreciating the manuscript and grasping the essence and the detail of the STID vision.

We understand that clear examples of rehabilitation care for Non-Communicable Diseases, and of innovative technologies as optimized through the STID model to target them, would have given a neat picture of the model outcomes.

However, our main focus was not on specific technologies/platforms. Rather, it was on the processes that the STID model enables, in order to integrate the standpoints (action and attitudes) of the various stakeholders involved. On a larger scale, the paper targets the feasibility testing of the STID model as a tool in a governance strategy aimed at tuning the agenda setting of innovation in rehabilitation care through a participatory methodology.

All digital solutions proposed, developed and tested with the STID model, are designed to be used in rehabilitation settings and to deliver services remotely, exploiting the potential of digital tools.

Although in clinical practice there are services in which patients carry out their rehabilitation sessions in groups, with a single health professional supervising the patients at the same time, establishing a “1 to many” ratio, STID technologies envisage establishing a 1 to 1, doctor/healthcare professional-patient ratio, with a closed loop approach. In this way, each healthcare professional presides over and monitors the sessions and rehabilitation pathway of a single patient at a time, both with a synchronous and an asynchronous approach.

This dynamic occurs for two reasons: while on the one hand there is an issue concerning the usability of technologies, by healthcare professionals and by patients, on the other hand the Italian regulatory context does not provide the reimbursement for rehabilitation approaches that are not 1:1. This represents a major disincentive for Innovators and Public Stakeholders to test and develop different systems.

So, thank you again for highlighting that such a pivotal aspect of the manuscript (focus on processes and not on specific technologies) had not clearly been put in focus. We added it in the Introduction section at the beginning of the manuscript, just after detailing the knowledge gap that the study presented in the paper aims to address. And we also hope that the information about the technologies designed, though not explored in the paper, will comply with your request.

Reviewer 2 Report

For science , the new patient-citizen perspective is important for the development and introduction of new technologies in rehabilitation care for Noncommunicable Diseases (NCDs).

The new ICT-based Smart&Touch-ID operating model supports the validation of rehabilitation with technology for people with disabilities.

To increase the quality of the manuscript, the authors should explain in detail the following aspects:

1. Expand the basic criteria to guarantee the economic sustainability of the model, and clarify whether it includes the rehabilitation process. How the needs and the territory influence.

2. List the multidimensional aspects of technological evaluation (clinical, economic and well-being) for the co-design and/or co-development of ICT-based solutions. Clarify what are innovative technologies?

3. Expand the needs analyzed in this study for rehabilitation for chronic disabilities.

4. In the operational framework, make the Smart&Touch-ID approach clear.

5. Show the endorsement of the ethics committee for the approval of the trial protocol with patients (95 participants).

6. Explain the variables used to measure health-related quality of life

7. Expand how to assess the impact of beliefs towards rehabilitation.

8. Specify what are the public health demands? in the context of the study carried out.

9. Review and update the following references:

to. #4 and #20: from 2006. Considered old

b. #19 from 1995: very old

c. #21 from 1992: very old

d. #27 from 1989: very old

and. #5 and #18: from 2010.

Author Response

We would like to thank the Reviewer for the interesting comments concerning our manuscript.

Here are our point-by-point answers to your comments.

  1. Expand the basic criteria to guarantee the economic sustainability of the model and clarify whether it includes the rehabilitation process. How the needs and the territory influence.
  2. List the multidimensional aspects of technological evaluation (clinical, economic and well-being) for the co-design and/or co-development of ICT-based solutions. Clarify what are innovative technologies?

By “innovative technologies” are meant the digital solutions for rehabilitation directed at people with Non-communicable Diseases (NCDs) that Innovators could incorporate into the STID flow through the participation to a Challenge. The operational flow of the STID model has been designed to accommodate and support the development and validation of both digital solutions in a concept stage and more mature solutions already represented by digital devices or solutions in a prototype form.

The operational flow of the STID model is described in Section 1.2 (Embedding patients’ perspectives into an operational framework for technological innovation in rehabilitation care: The Smart-Touch-ID approach) and pictured in Fig. 1. To better clarify this aspect, at the beginning of the manuscript (Section 1. Introduction) we stated that, although the STID model operates to optimize the co-design and development of innovative solutions for rehabilitation care for NCDs, in the present paper the focus is not on specific digital solutions as outputs of the STID multidimensional evaluation flow or on the multidimensional approach. The attention is rather drawn to the processes that the STID model enables, to integrate the standpoints (action and attitudes) of the various stakeholders to get to technology optimization and to create a co-design of the solutions by collecting the perception of citizens.

In general, the multidimensional model applied for the entire STID operational process provided for the integration of multiple aspects, which can influence the success of a technology used in NCD rehabilitation.

The evaluation process and support flow, in the development of rehabilitation technologies was quite complex and consisted of up to 8 validation moments. In each step several aspects and items were evaluated, consistent with the specific state of development of each digital solution when approaching the STID model.

The multidimensional assessment aspects included: the validation of the technology, that is, whether the technology worked for the intended clinical use; adherence to Smart, Touch and ID criteria (respectively, technological, well-being/engagement/usability, and clinical aspects); consistency with aspects of effective governance (aspects of economic sustainability and potentiality of inclusion in reimbursement pathways); compliance with international or EU standards (ISO or medical device); compliance with patent legislation and possible development of strategies to protect intellectual property.

In case of digital solutions with a more mature design or prototypes, a Health Technology Assessment was developed to assess their impact in the Italian healthcare context through the EUnetHTA Core Model approach, a multidimensional methodology that considers the introduction of innovations in healthcare from different perspectives, such as clinical, economic, organizational, social, ethical, legal and accessibility of care.

Within this view, in the manuscript, the Authors focused on aspects related to the “social” perspective and considering the role of the users’ point of view.

  1. Expand the needs analyzed in this study for rehabilitation for chronic disabilities.

We thank the reviewer for remarking this point. In section 1.2. (Embedding patients’ perspectives into an operational framework for technological innovation in rehabilitation care: The Smart-Touch-ID approach; p.4) we stated that the operational flow of the STID model originates from the launch a “Challenge” in response to a specific “need”. At p. 4 of the same section, we detailed that the evidence concerning needs is collected by different sources. In the top-down modality, it originates from domain experts and from clinicians, on the basis of the information they routinely collect from patients during the rehabilitation activities. In the bottom-up modality, patients and citizens communicate their experiences and attitudes through targeted questionnaires administered on the STID site (waves). The focus of the present manuscript is on how patients’ and citizens’ experiences and attitudes on rehabilitation care are embedded into the operational flow of the STID model, providing preliminary evidence for its feasibility under this specific regard.

Considering that an essential role for the co-design and investigation of the citizens point of view was represented by the possibility of having ad adequate picture of the different visions and perceptions of the digital solutions’ potential users, it was decided to detect this information through multiple data source. The triangulation of sources is a fundamental approach in scientific research, as it allows increasing the validity and accuracy of the results obtained. The use of multiple data sources makes it possible to integrate different perspectives and identify possible discrepancies, thus improving the completeness and robustness of the conclusions reached. Furthermore, the triangulation of sources makes it possible to overcome any limitations or flaws of individual sources, thus ensuring greater reliability of results. In summary, the use of adequate triangulation of sources is a good practice in scientific research, as it allows for more accurate and reliable results (Denzin et al., 1978).

(Denzin, N. K. (1978). Triangulation: A Case for Methodological Evaluation and Combination. Sociological Methods, 339-357).

  1. In the operational framework, make the Smart&Touch-ID approach clear.

 Thank you very much for your comment. The description of the operational flow has been made clearer in paragraph 1.2. (Embedding patients’ perspectives into an operational framework for technological innovation in rehabilitation care: The Smart-Touch-ID approach).

  1. Show the endorsement of the ethics committee for the approval of the trial protocol with patients (95 participants).

The Legal Office of the University of Milano-Bicocca, Data Controller and Data Processor for the questionnaire on patients’ and citizens’ experiences about rehabilitation, to be administered during the feasibility study reported in the present manuscript, inspected all relevant content (questions, purpose, administration modality, data processing and storage).

Since the link to access the questionnaire was not distributed directly to participants (through email or other devices that could identify the potential participant), but only through news in newsletters and on the web-site, and since data were anonymous and without any possibility of tracing the identity of the respondents, the Legal Office resolution was that participants had to necessarily read an information note on the “disclosure for the processing of data” before compiling the questionnaire and giving the consent to participate.

That is, based on the information communicated and the nature of the questions administered, the  Data Controller and Data Processor opinion was that a specific ad hoc information notice should be prepared for the research project in question and administered upstream, before proceeding with the completion of the questionnaire. For this reason, we were only required to prepare an information notice. This information notice was prepared on the basis of the template given by the University and its content validated by the University Legal Office.

On the STID website, this step was made mandatory to commence questionnaire compilation. No approval from the Ethics Committee was considered mandatory.

  1. Explain the variables used to measure health-related quality of life

The perceived health-related Quality of Life of respondents with chronic disabilities was measured through the WHODAS 2.0 scale. The reviewer can find this scale's detailed description and reference in section 2.2.2.

  1. Expand how to assess the impact of beliefs towards rehabilitation.

Beliefs towards rehabilitation impact on the flow of development of technologies and on the design of user interfaces. They are detected (not assessed) through the STID flow. Rather, they are turned into prompts for Innovators to optimize the design and development of technologies for rehabilitation care for NCDs. Beliefs are not used as data in itself, but as perceptions useful to modify or justify the accessibility, the adequacy (e.g., with respect to health and well-being needs) and the perceived ease of use of the digital solutions characteristics and interfaces.

  1. Specify what are the public health demands? in the context of the study carried out.

Thank you very much for your kind suggestion. In this moment the public health supply of rehabilitation treatments for people with Non-Communicable diseases is not able to cover the populations needs. This misalignment between supply and demand depends on the one hand, by the inability to guarantee adequate services to all patients in need; and, on the other hand, by the difficulty to fully exploit the potential of technology to innovate rehabilitation.  Low capacity and delayed access are also due to outdated and poorly updated reimbursement systems and tariffs, not in line with the current technological possibilities.

This aspect may also discourage Innovators since they cannot find adequate pricing in relation to their purpose to explore new approaches and/or solutions.

The STID model, combining patients’ and citizens’ perspectives with the Governance dimension, would support the public service to solve this specific problem, not only developing the most promising solutions for the market, but also providing clear indications of how they can be reimbursed by the healthcare service.

  1. Review and update the following references:
  2. #4 and #20: from 2006. Considered old
  3. #19 from 1995: very old
  4. #21 from 1992: very old
  5. #27 from 1989: very old

and. #5 and #18: from 2010.

The following references are related to scale validation and statistical approaches used for data analysis; therefore, they are not recent, but there is no reason to assume they are old.

  1. Ustün, T.B.; Chatterji, S.; Kostanjsek, N.; Rehm, J.; Kennedy, C.; Epping-Jordan, J.; Saxena, S.; von Korff, M.; Pull, C.; WHO/NIH Joint Project Developing the World Health Organization Disability Assessment Schedule 2.0. Bull World Health Organ 2010, 88, 815–823, doi:10.2471/BLT.09.067231. (VALIDATION OF THE WHODAS 2.0)
  2. West, S.G.; Finch, J.F.; Curran, P.J. Structural Equation Models with Nonnormal Variables: Problems and Remedies. In Structural equation modeling: Concepts, issues, and applications; Sage Publications, Inc: Thousand Oaks, CA, US, 1995; pp. 56–75 ISBN 978-0-8039-5317-8. (STATISTICAL APPROACH)
  3. Worthington, R.L.; Whittaker, T.A. Scale Development Research: A Content Analysis and Recommendations for Best Practices. The Counseling Psychologist 2006, 34, 806–838, doi:10.1177/0011000006288127. (STATISTICAL APPROACH)
  4. Comrey, A.L.; Lee, H.B. Interpretation and Application of Factor Analytic Results. In A First Course in Factor Analysis; Psychology Press, 1992 ISBN 978-1-315-82750-6. (STATISTICAL APPROACH)

Reference 5 is from 2019, not 2010

  1. Topol, E.J. A Decade of Digital Medicine Innovation. Sci Transl Med 2019, 11, eaaw7610, doi:10.1126/scitranslmed.aaw7610.

Reference 4 was deleted. Three more recent references were inserted as substitutes:

Ellis TD, Colón-Semenza C, DeAngelis TR, Thomas CA, Hilaire MS, Earhart GM, Dibble LE. Evidence for Early and Regular Physical Therapy and Exercise in Parkinson's Disease. Semin Neurol. 2021 Apr;41(2):189-205. doi: 10.1055/s-0041-1725133. Epub 2021 Mar 19. PMID: 33742432; PMCID: PMC8678920.

Jelcic N, Agostini M, Meneghello F, Bussè C, Parise S, Galano A, Tonin P, Dam M, Cagnin A. Feasibility and efficacy of cognitive telerehabilitation in early Alzheimer's disease: a pilot study. Clin Interv Aging. 2014 Sep 24;9:1605-11. doi: 10.2147/CIA.S68145. PMID: 25284993; PMCID: PMC4181448.

Ryrsø CK, Godtfredsen NS, Kofod LM, Lavesen M, Mogensen L, Tobberup R, Farver-Vestergaard I, Callesen HE, Tendal B, Lange P, Iepsen UW. Lower mortality after early supervised pulmonary rehabilitation following COPD-exacerbations: a systematic review and meta-analysis. BMC Pulm Med. 2018 Sep 15;18(1):154. doi: 10.1186/s12890-018-0718-1. PMID: 30219047; PMCID: PMC6139159

Reference 27 was deleted.

Reviewer 3 Report

1. The introduction is too long and divided into two subheadings. This makes it difficult for me to figure out what research gap the authors are trying to fill.

2. I think section 2.3 can be included in section 3 "Results".

3. Since the authors have done the EFA analysis, please provide a detailed table of the results of the efa analysis.

Author Response

We would like to thank the Reviewer for the stimulating comments on the manuscript.

Here are the point-by-point answers to your comments.

  1. The introduction is too long and divided into two subheadings. This makes it difficult for me to figure out what research gap the authors are trying to fill.

 We thank you for highlighting this issue.

At the beginning of the manuscript (section 1. Introduction) we detailed the gap that the paper addresses, the aim of the study presented, and briefly summarized the subsequent sections.

  1. I think section 2.3 can be included in section 3 "Results".

The structure of the manuscript is typical of scientific articles. Indeed, Section 2 describes the method and procedure of the study, while Section 3 describes the study's results. Section 2.3 describes the statistical techniques adopted to analyze the data. Therefore, it belongs to the section describing the method and not to the results. We trust that the structure of the current sections is consistent with what is typically found in scientific articles and that it is clear to readers.

  1. Since the authors have done the EFA analysis, please provide a detailed table of the results of the efa analysis.

Following the reviewer's request, we added a table describing factor loadings from the EFA about the "use frequency of technologies" scale (Table 2) and a table describing factor loadings from the EFA about the "proficiency in the technologies use" scale (Table 4). In this way, we have reported in detail all the results of the EFAs.

Reviewer 4 Report

Since 1990, the need for rehabilitation for people with Non-Communicable Diseases (NCDs) has increased to involve up to 2.41 billion people.

However, rehabilitation is still a very specialized service, mostly directed at severe disabilities and cannot guarantee intensive access to people in the early to medium stages of the disease, when timed interventions can slow-down the progression of the disease. Remote rehabilitation care through telecommunication and information technologies is the ideal candidate to reach all people with NCDs in need.

The authors proposed with this aim  the Smart&Touch-ID (STID) operational model  designed and developed to support the creation and validation of tech-enhanced rehabilitation pathways for people with chronic disabilities

The model vision aims to harmonize the health (ID) and well-being (Touch) needs of patients-citizens with the design and development (SMART) of technological solutions, guaranteeing their economic sustainability. Based on patients’ needs, the STID flow incorporates the patient perspective in the technology assessment, and enables the co-design of technological solutions with a multi-stakeholder approach and multidimensional evaluation tools.

The authors  present preliminary evidence on the rehabilitation needs for chronic disabilities and illustrate how they operate into the model, enabling the creation and stabilization of shared practices for technological innovation in rehabilitation care.

Interesting study.

I have some minor suggestions.

1.      The abstract must be improved better summarizing the sections.

2.      Better connect the two paragraphs of the introduction.

3.      Avoid the use of WE.

4.      The web portals must be inserted as links.

5.      Describe figure 1

6.      Insert a clear purpose

7.      Insert the limitations in the discussion

Author Response

We thank the reviewer for appreciating the manuscript. We have amended the manuscript following point by point all the suggestions made.

  1. The abstract must be improved better summarizing the sections.

The Abstract was substantially revised including a clear purpose.

  1. Better connect the two paragraphs of the introduction

At the beginning of the manuscript (section 1. Introduction), a few paragraphs were added that detail the gap that the paper addresses, the aim of the study presented, and briefly summarize the subsequent sections. This way, the two paragraphs of the Introduction are connected by the conceptual flow illustrated at the beginning of the manuscript.

  1.          Avoid the use of WE

Following the reviewer's suggestion, we revised the text using the impersonal.

  1. The web portals must be inserted as links.

The link to the STID web portal is shown in section 1.2 (Embedding patients’ perspectives into an operational framework for technological innovation in rehabilitation care: The Smart-Touch-ID approach; p.5).

  1. Describe figure 1

We have modified the figure to be more compliant with the description.

  1. Insert a clear purpose

At the beginning of the manuscript (section 1. Introduction) we stated that the purpose of the paper is to illustrate how the Smart&TouchID (STID) model addresses the need to incorporate patients’ evaluations into a multidimensional technology assessment framework by presenting a feasibility study of model application with regard to rehabilitation experiences of people living with NCDs.

  1. Insert the limitations in the discussion

Thank you for highlighting this issue. The main limitation of the STID model was inserted in the manuscript. 

The main limitation of the STID model is related to the uncertainty of the real possibility to implement innovation in rehabilitation care process, not depending directly on the STID model, but on the decision-making process taking place at a regulatory level.

In this moment the public health supply of rehabilitation treatments for people with Non-Communicable diseases is not able to cover the populations needs. This misalignment between supply and demand depends on the one hand, by the inability to guarantee adequate services to all patients in need; and, on the other hand, by the difficulty to fully exploit the potential of technology to innovate rehabilitation.  Low capacity and delayed access are also due to outdated and poorly updated reimbursement systems and tariffs, not in line with the current technological possibilities.

This aspect may also discourage Innovators since they cannot find adequate pricing in relation to their purpose to explore new approaches and/or solutions.

The STID model, combining patients’ and citizens’ perspectives with the Governance dimension, would support the public service to solve this specific problem, not only developing the most promising solutions for the market, but also providing clear indications of how they can be reimbursed by the healthcare service.

Reviewer 5 Report

Thank you for the opportunity to review this manuscript.

I find the introduction and the discussion sections very confusing and unclear.  This paper reads much more like a technological/innovation promotion versus an investigation into the use of the SMART STID equipment and methods.  Healthcare journal readers and medical providers will require a clear, basic understanding of this technology at the user-end level in order to make a personal decision if acceptable or not, in conjunction with the study's quantitative results.

The STID technology and uses are not clear.  It is suggested that the authors provide suggestions and/or examples (case discussions) to show the end-user perspective of this technology.  A good time to do this would be the figure with a circular model.

I do not understand the patient-citizen term/concept.  Where does the citizen part fall in?  This needs to be clarified.

There is a survey involved in this study with human subjects, yet no IRB approval.

Author Response

Reviewer 5

Thank you for the opportunity to review this manuscript.

I find the introduction and the discussion sections very confusing and unclear.

This paper reads much more like a technological/innovation promotion versus an investigation into the use of the SMART STID equipment and methods.  Healthcare journal readers and medical providers will require a clear, basic understanding of this technology at the user-end level in order to make a personal decision if acceptable or not, in conjunction with the study's quantitative results.

The STID technology and uses are not clear.  It is suggested that the authors provide suggestions and/or examples (case discussions) to show the end-user perspective of this technology.  A good time to do this would be the figure with a circular model.

I do not understand the patient-citizen term/concept.  Where does the citizen part fall in?  This needs to be clarified.

There is a survey involved in this study with human subjects, yet no IRB approval.

Reply

We appreciate the reviewer’s suggestion, and modified the manuscript making it clear that STID is not a technology, but a model that operates to optimize the co-design and development of innovative solutions for rehabilitation care for NCDs.

We understand that clear examples of rehabilitation care for NCDs, and of innovative technologies as optimized through the STID model to target them, would have given a neat picture of the model outcomes.

However, our main focus was not on specific technologies as outcomes of the STID multidimensional evaluation flow. Rather, the attention is drawn to the processes that the STID model enables to integrate the standpoints (action and attitudes) of the various stakeholders to get to technology optimization. On a larger scale, the paper targets the feasibility testing of the STID model as a tool in a governance strategy aimed at tuning the agenda setting of innovation in rehabilitation care through a participatory methodology.  This way, the ambition of the STID model is to transform the way in which stakeholders in rehabilitation care (i.e. Innovators, Clinicians, Patient-Citizens) can cooperate to design and develop innovative technologies for rehabilitation for NCDs.

We detailed these aspects at the beginning of 1. Introduction.

Accordingly, the choice to refer to “patients-citizens” reflects this vision, whereby patients are not only seen as “people suffering from a clinical condition”, but also as active stakeholders whose perspectives can impact - and, in the STID flow, do impact - through their experiences and attitudes, on the design and development of technological solutions in order to make rehabilitation a widespread service. This impact, in STID, is modeled not as an isolated step or as an endpoint in technology development but is integrated since the very first steps of the development process.

Finally, as regards the IRB approval, we followed the line suggested by the Legal Office of the University of Milano-Bicocca, Data Controller and Data Processor for the questionnaire on patients’ and citizens’ experiences about rehabilitation, to be administered during the feasibility study reported in the present manuscript.

The DPO inspected all relevant content (questions, purpose, administration modality, data processing and storage). Since the link to access the questionnaire was not distributed directly to participants (through email or other devices that could identify the potential participant), but only through news in newsletters and on the web-site, and since data were anonymous and without any possibility of tracing the identity of the respondents, the Legal Office resolution was that participants had to necessarily read an information note on the “disclosure for the processing of data” before compiling the questionnaire and giving the consent to participate.

That is, based on the information communicated and the nature of the questions administered, the  Data Controller and Data Processor opinion was that a specific ad hoc information notice should be prepared for the research project in question and administered upstream, before proceeding with the completion of the questionnaire. For this reason, we were only required to prepare an information notice. This information notice was prepared on the basis of the template given by the University and its content validated by the University Legal Office.

On the STID website, this step was made mandatory to commence questionnaire compilation. No approval from the Ethics Committee was considered mandatory.

Round 2

Reviewer 5 Report

Thank you for making these revisions.  Figure 1 certainly helps and the additional discussion surrounding the model itself is very beneficial.

Clarify has come a long way throughout as well.  Thank you for the additional comments on the IRB approval not being needed per your institution's guidance/etc.  That works for me.